# Measurement of 3D Wrist Angles by Combining Textile Stretch Sensors and AI Algorithm

**DOI:** 10.3390/s24051685

**Published:** 2024-03-05

**Authors:** Jae-Ha Kim, Bon-Hak Koo, Sang-Un Kim, Joo-Yong Kim

**Affiliations:** 1Department of Materials Science and Engineering, Soongsil University, Seoul 156-743, Republic of Korea; kim76erss@gmail.com (J.-H.K.); soongsilkoo@soongsil.ac.kr (B.-H.K.); 2Department of Smartwearable Engineering, Soongsil University, Seoul 156-743, Republic of Korea; tkddnsl0723@naver.com

**Keywords:** smart wearable sensors, wrist angle, textile stretch sensors, multi-layer perceptron, AI algorithms

## Abstract

The wrist is one of the most complex joints in our body, composed of eight bones. Therefore, measuring the angles of this intricate wrist movement can prove valuable in various fields such as sports analysis and rehabilitation. Textile stretch sensors can be easily produced by immersing an E-band in a SWCNT solution. The lightweight, cost-effective, and reproducible nature of textile stretch sensors makes them well suited for practical applications in clothing. In this paper, wrist angles were measured by attaching textile stretch sensors to an arm sleeve. Three sensors were utilized to measure all three axes of the wrist. Additionally, sensor precision was heightened through the utilization of the Multi-Layer Perceptron (MLP) technique, a subtype of deep learning. Rather than fixing the measurement values of each sensor to specific axes, we created an algorithm utilizing the coupling between sensors, allowing the measurement of wrist angles in three dimensions. Using this algorithm, the error angle of wrist angles measured with textile stretch sensors could be measured at less than 4.5°. This demonstrated higher accuracy compared to other soft sensors available for measuring wrist angles.

## 1. Introduction

Internet of Things (IoT) and sensor technologies have advanced, deeply integrating into our daily lives. Particularly, in the fields of rehabilitation and sports, utilizing IoT devices to obtain precise joint motion data has become crucial [1]. For instance, in sports like badminton, achieving the correct wrist angle is essential for optimal performance, and providing feedback on wrist motion can enhance athletic abilities [2]. Similarly, sports such as basketball, tennis, and golf, where the wrist is frequently used, can benefit from preventing injuries associated with excessive wrist angles and improving performance based on proper wrist postures [3,4]. The wrist, comprising eight bones, is one of the most complex joints in the human body, involving six movements with a total of 3 degrees of freedom (DOF), including flexion–extension, ulnar–radial deviation, and pronation–supination. These movements can be combined in complex ways. Thus, accurately modeling this intricate structure requires complex equations and calculations, making the systems for measuring wrist angles complex [5]. 

Traditional systems used for wrist angle measurement can be categorized into three types: camera-based motion capture systems, rigid sensors, and soft sensors. Camera-based systems capture high-speed wrist movements using cameras and analyze them through algorithms. However, these systems require cameras to be positioned around the test area, limiting wrist movements to a confined space and reducing measurement speed due to complex image processing [6,7,8]. The second type involves attaching Inertial Measurement Units (IMUs), which are rigid sensors, to the wrist to measure motion through changes in acceleration and inertia. While free from sound and light interference, IMU sensors suffer from integration drift and the need for continuous absolute position readjustment [9]. To address these limitations, soft sensors like Dielectric Elastomer Sensors (DES) have been explored. DES have advantages such as high performance, lightweight, low operational noise, and cost-effectiveness. However, various drawbacks arise due to the properties of the dielectric materials used [10]. As a first example, acrylate elastomers exhibit excellent dielectric properties, such as large deformation and high breakdown strength. Nevertheless, the large pre-strain introduces practical issues, including stress relaxation and increased equipment complexity. Additionally, the low electrical-mechanical speed response and consequent low conversion efficiency result from the viscoelastic properties of acrylates. Their adaptability to environmental conditions is also inferior compared to other dielectric materials [11,12,13,14]. In the case of polyurethane (PU) elastomers, which represent another dielectric material used in DES, their complex network structure leads to a high Young’s modulus, limiting further development due to imposed constraints [15,16,17]. Lastly, widely used silicone elastomers have a lower dielectric constant than the previously mentioned dielectric materials. Consequently, silicones require higher driving voltages, posing risks to users and potentially causing critical damage to machinery. Moreover, they do not offer advantages in terms of destructive strength [18,19,20,21]. In this study, a stretch sensor, incorporating Single-Walled Carbon Nanotubes (SWCNT) into an E-band, was developed to address the limitations of previous soft sensors. This method, using stretch sensors, offers higher stability than DES, simplicity in fabrication, high reproducibility, cost-effectiveness, and the ability to adjust dimensions freely [22,23,24]. Furthermore, it has the advantage of precisely monitoring complex and rapid human movements due to its excellent durability, fast response, and low creep [25,26,27]. The stretch sensor, leveraging the elasticity of fabric, maintains initial resistance and changes resistance based on stretching. 

The method of analyzing wrist movement data using the existing soft sensor involves measuring only in fixed postures or using a decoupling method to eliminate coupling generated by the sensor. However, this approach has limitations in measuring free wrist movements in three dimensions, is not intuitive, and requires a separate and time-consuming calculation [8,28]. To overcome this, this paper introduces the use of Multi-Layer-Perceptron (MLP), a deep learning technique, for obtaining and analyzing wrist angle data, allowing for dynamic and unrestricted measurement of wrist angles. Furthermore, by directly utilizing MLP, coupling was not avoided but rather incorporated, leading to the creation of an intuitive model that does not require separate calculations.

## 2. Materials and Methods

### 2.1. Principle of Stretch Sensors

Stretch sensors operate based on the characteristic of resistance changing with the elongation of an object. The principle of stretch sensors is as follows: When the length (L) of the object increases due to external forces (i.e., L = L_0_ + ∆L), the cross-sectional area decreases, leading to an increase in resistance. Conversely, if the length decreases, the cross-sectional area increases, resulting in a decrease in resistance. Thus, when the fabric is stretched by an external force, both the positive piezoresistive effect (an increase in resistance as the fabric expands) and the negative piezoresistive effect (a decrease in resistance as the fabric stretches) occur simultaneously, allowing for the measurement of resistance changes in response to length variations.
R = R_0_ × (1 + GF × ε)(1)

In Equation (1), R represents the current resistance value of the stretch sensor, while R_0_ denotes the initial resistance value of the stretch sensor. GF stands for the gauge factor, which indicates the sensitivity of the stretch sensor. ε = ΔL/L represents the strain (deformation) applied to the object, where ΔL is the change in length, and L is the original length. In this way, the stretch sensor can precisely measure the changes in resistance that occur with changes in the length of the object, thereby accurately determining the extension and deformation of the object. See Figure 1.

### 2.2. Textile Stretch Sensor Fabrication

The stretch sensor used in this study was fabricated as follows. A method involving the immersion of a fabric in a 0.1wt% water-dispersed SWCNT (Single-Wall Carbon Nanotube, KH WAR 1200ST, KH CHEMICALS, Zwijndrecht, Netherlands) solution onto an E-band (PET(80%)/SP(20%), KOLON Co., Ltd., Seoul, Republic of Korea) was employed. The SWCNT is a cylindrical carbon nanostructure known for its excellent conductivity and is commonly employed in the fabrication of conductive sensors from textiles. First, the E-band in the SWCNT dispersion was immersed for 1 min. Afterwards, to remove the moisture present in the E-band, it was padded using a vertical padder machine (DL-2500V, DAELIM lab, Siheung-si, Republic of Korea) and finally dried for 10 min at a temperature of 100 °C using a two-way drying machine (DLS-6600, DAELIM lab). Furthermore, the padding machine plays a role in evenly dispersing the SWCNT solution across the E-band, enhancing the uniformity of the produced stretch sensor. The E-band, after the immersion process, was then cut using a laser cutting machine (JQ9060, JinQiang, Quanzhou, China) into dimensions of 90 × 10 (mm), completing the fabrication of a stretch sensor capable of measuring wrist angles [29]. The immersion process is the simplest and most efficient method for dispersing conductive materials into fabric [30]. Figure 2 illustrates the fabrication process of stretch sensors.

### 2.3. Segmentation of Wrist Joint Movement

Wrist exercises involve complex movements that engage muscles and the skeletal system. Types of wrist exercises include flexion/extension (F/E), radial/ulnar deviation (R/U), and pronation/supination (P/S). The range of motion for each is as follows: F/E ranges from 70 to 90 degrees, R ranges from 15 to 25 degrees, U ranges from 20 to 45 degrees, and P/S ranges from 80 to 90 degrees. In Figure 3, three-axis movements related to wrist exercises are illustrated.

### 2.4. The Attachment Location of Stretch Sensors 

The precise placement of sensors at the intended measurement locations is crucial for various reasons. Firstly, it directly impacts the accuracy and reliability of the measured data, ensuring that the obtained results align with the true conditions. Secondly, accurate sensor placement enhances consistency, leading to more reliable and repeatable measurements under the same conditions. Thirdly, proper sensor placement minimizes interference and mutual effects when multiple sensors are employed, contributing to cleaner and more accurate data. Lastly, positioning sensors at their intended locations allows for optimal utilization of each sensor’s designed characteristics, ensuring they perform at their best. Therefore, careful consideration and implementation of accurate sensor placement are vital for obtaining meaningful and trustworthy measurement outcomes. 

The back of the hand undergoes significant movement during wrist flexion and extension, making it an ideal location for capturing the range of motion of the wrist joint. Placing the sensor on the dorsum of the hand reduces interference from other muscle movements or joint actions, allowing for a specific focus on measuring the flexion and extension angles of the wrist. The accessibility of the back of the hand makes it convenient for sensor attachment, ensuring secure and stable placement for reliable measurements during various activities. Additionally, the attachment on the back of the hand aligns well with the extensor tendons and muscles involved in wrist extension. By attaching the stretch sensor on the back of the hand, accurate and reliable measurements of wrist flexion and extension can be obtained. To measure ulnar and radial deviation angles, it is necessary to attach the sensor to the side of the wrist. Anatomically, the wrist angle tends to have a greater magnitude during ulnar deviation compared to radial deviation. Therefore, positioning the sensor closer to the radius bone rather than the ulna bone may yield better results, depending on the characteristics of the stretch sensor. Pronation and supination involve rotation driven by the pronator teres and pronator quadratus muscles. The area of skin above these muscles stretches during this motion, making it a suitable location for attaching the stretch sensor to obtain sensor values. Specifically, due to significant interference during flexion–extension, the optimal position for measuring pronation–supination angles is the pronator teres. In Figure 4, the depicted positions represent the actual sensor attachment locations, indicating the types of movements measurable at those locations. Figure 5 illustrates the actual attachment locations of stretch sensors, referencing Figure 4.

### 2.5. The Method of Obtaining Data for Stretch Sensors and IMU 

In Figure 6, the illustration depicts the method of collecting data from sensors and the attachment positions. To obtain data from stretch sensors, Arduino Uno was employed. Figure 7 depicts the circuit diagram illustrating the connection between a stretch sensor and an Arduino Uno. The input voltage is 5 V, and a 220 KΩ resistor has been utilized. The wire connecting the stretch sensor and the breadboard in Figure 7 was actually connected using alligator clips. The applied 5 V voltage from Arduino was divided into values ranging from 0 to 1023, converting the analog signal into data signals. As the stretch sensor expands, the resistance increases, leading to an increase in data values. Conversely, when the stretch sensor contracts, the resistance decreases, resulting in a decrease in data values. Through an IMU, it is possible to determine the angles of the wrist in three axes: flexion–extension, radial–ulnar deviation, and pronation–supination (XYZ axes). This information can be utilized as reference angles for obtaining the angles with a stretch sensor. The data obtained through Arduino and IMU can both be digitally stored on a computer. Later, the data can be used to enhance the accuracy of the sensors through the upcoming MLP (Multi-Layer Perceptron).

### 2.6. Multi-Layer Perceptron (MLP)

To compare the data obtained from stretch sensors with the IMU data used for actual wrist angle measurement, we utilized a deep learning technique known as Multi-Layer Perceptron (MLP). Unlike traditional analysis methods, deep learning can learn features directly from raw data without significant preprocessing, efficiently learning patterns to improve recognition accuracy. This advantage allows for the collection of data directly at the angle when the wrist angle is three-dimensionally fixed or obtained as a virtual result through a regression equation. However, to prevent overfitting, a considerable amount of data and GPU devices for accelerating complex computations are required [31]. Deep learning is increasingly employed in extracting and classifying features of complex human activities from mobile and wearable sensors. In this study, we propose a deep learning approach for wrist angle measurement based on stretch sensors. MLP, one of the deep learning technologies, is a type of artificial neural network with a structure consisting of input, hidden, and output layers. MLP uses the backpropagation algorithm to adjust weights and minimize errors, enabling the model to adapt to the learned data. In this paper, the MLP model incorporated 1000 training epochs, a learning rate of 0.1, and 10 neurons in the hidden layer. The training dataset constituted 70% of the data, while 15% was used for validation, and the remaining 15% for testing. To prevent overfitting, it is necessary to use only a portion of the training data. Overfitting refers to the phenomenon where the model becomes too tailored to the training data, making it difficult to generalize to other data. Test data are employed to evaluate the actual performance of the model. Additionally, validation data are used to assess and adjust the model’s performance during training. This process helps enhance the model’s generalization performance. Hyperparameters are adjusted based on experimentation to achieve the optimum R values [32,33,34]. Additionally, the non-linearity feature of MLP allows it to grasp complex patterns in wrist movements, revealing a variety of patterns compared to linear models [35]. In this research, stretch sensor values were input into the input layer, and wrist angles obtained from IMU were input into the output layer to train the MLP. This process allowed the recognition of patterns between the input and output layers, teaching the stretch sensor algorithm for wrist angles. See Figure 8.

## 3. Results

To evaluate the performance of the stretch sensor and the enhanced accuracy achieved through MLP, we conducted a performance comparison with traditional linear regression analysis using the same dataset in Section 3.1 and Section 3.2. In Section 3.4 and Section 3.5, we utilized the 3-axis angle measurement algorithm created through MLP to measure angles during actual movements and compared the errors between the measured angles and the actual angles. 

### 3.1. Linear Regression Analysis

In this study, participants wore a stretch sensor and performed wrist movements along three axes. The neutral position was based on 0 degrees. For flexion/extension (F/E), (-) represented extension, and (+) represented flexion. For radial/ulnar deviation (R/U), (−) indicated radial deviation, and (+) indicated ulnar deviation. Finally, for pronation/supination (P/S), (−) denoted pronation, and (+) denoted supination angles. Regarding the measurement results, the wrist angles for F/E, R/U, and P/S were varied by 20 degrees, 10 degrees, and 20 degrees, respectively. The average output voltage and the average angle during these intervals were calculated. Subsequently, linear regression analysis was performed for fitting. The relationship between wrist angles measured using an IMU (Inertial Measurement Unit) as a reference and the output voltage of the stretch sensor is illustrated in Figure 9. 

The coefficient of determination, R, is a statistical measure representing the proportion of variance in the dependent variable that can be explained by the independent variable. It ranges from 0 to 1, with higher values indicating a better fit of the model to the data. The R values obtained from linear regression analysis for the sensor were 0.8874, 0.8989, and 0.9421 for F/E, R/U, and P/S, respectively. 

### 3.2. MLP Regression Analysis

To address the unsatisfactory R value, the application of MLP for non-linear regression analysis was employed, resulting in a significant improvement in the numerical value, and the resulting outcomes are presented in Figure 10. The utilized MLP model involved 1000 training epochs, a learning rate of 0.1, and 10 neurons in the hidden layer. The training dataset constituted 70%, while 15% was used for validation and the remaining 15% for testing.

Figure 10 presents a compelling demonstration of the enhancement in sensor performance, achieved by utilizing non-linear regression analysis carried out through Multi-Layer Perceptron (MLP). Moreover, it is noteworthy that, unlike Figure 9, the values have improved even when utilizing unfiltered raw data. In this analysis, the R values, which are indicative of the correlation between the variables, showed an obvious improvement across all three sensors in question. The F/E sensor, which initially had an R value of 0.8874, showed a significant increase, reaching 0.97014. The R/U sensor too, exhibited a similar pattern of improvement, with its R value escalating from 0.8989 to 0.95644. Lastly, the P/S sensor saw its R value climb from 0.9421 to 0.97165. On examining Figure 10a,c,e, it was observed that the values of the stretch sensors, during the actual execution of wrist exercises, displayed a non-linear and irregular pattern of increase. This pattern is not consistent and features a degree of unpredictability, making it difficult to analyze using traditional methods. However, Figure 10 effectively demonstrated that despite this unpredictability, the use of MLP can indeed enhance the performance of such non-linear and irregular sensors. This enhancement is primarily due to MLP’s ability to model complex relationships and capture the inherent non-linearity in the sensor data. Therefore, the use of MLP offers a significant advancement in utilizing non-linear and irregular sensors more effectively and accurately, thereby improving the overall sensor performance.

### 3.3. Coupling between the Each Motion

This study extensively examines a method for accurately measuring the three-dimensional angles of the wrist. To achieve this, it is crucial to effectively address the ‘Coupling’ issue associated with the sensors used during wrist movements. Previous studies have adopted a decoupling formula to remove coupling, indirectly resolving the problem. However, in this study, a novel approach is proposed that directly utilizes coupling during wrist movements to accurately measure the three-dimensional angles of the wrist. This method offers a new perspective by considering coupling not as a problem but as a solution. Furthermore, the accuracy is improved by directly inputting the values from three sensors into a Multi-Layer Perceptron (MLP) and obtaining the corresponding angles as output, without the need for a separate decoupling algorithm. This approach enables rapid acquisition of results. Figure 11 visualizes the degree of coupling generated by a single-axis sensor in relation to other sensors. This visualization provides a clearer understanding of the interaction between sensors and their impact. To achieve this, voltage changes were applied to compare different sensor values.

Figure 11 shows the coupling level of sensors generated during wrist movements. It provides information on how the values of two other sensors attached to the wrist measurement device change, in addition to the sensor that measures the angle during wrist movements. (a) Corresponds to the ‘pitch’ of the gyro sensor, representing the appearance during F/E movements. It demonstrates that coupling primarily occurs in the R/U sensor during pitch movements. (b) Corresponds to the ‘yaw’ of the gyro sensor, representing the appearance during R/U movements. It exhibits a significant coupling value in the F/E sensor during roll movements. (c) Corresponds to the ‘roll’ of the gyro sensor, representing the appearance during P/S movements. It shows that there is a significant coupling level mainly in the F/E sensor during yaw movements.

### 3.4. Three-Dimensional Wrist Angle Measurement Algorithm

Figure 11 provides an overview of the existence of coupling during wrist movements. In this study, the ultimate goal was to measure wrist angles freely in three dimensions. This was achieved by directly incorporating the coupling values into the MLP algorithm. In other words, the voltage values of the three sensors were used as input, and the corresponding angles were included as output values. This enabled obtaining wrist angle values in three dimensions by obtaining values for all three axes. 

The R values in Figure 12 serve as a measure of the discrepancy between the actual angles and the values generated by the MLP model algorithm. The R values indicate how well the model fits the data and how closely its predictions align with the ground truth angles in three dimensions. The MLP model was trained with specific hyperparameters, including 1000 training epochs, a learning rate of 0.1, and 10 neurons in the hidden layer. The training dataset accounted for 70% of the data, while 15% was allocated for validation, and the remaining 15% was used for testing. The R values for train, validation, and test are 0.93972, 0.94826, and 0.94742, respectively. This indicates that the model is well fitted for measuring wrist angles in three dimensions.

### 3.5. Example of the 3D Wrist Angle Measurement Algorithm

To assess the accuracy of the 3D wrist angle measurement model, a set of voltage values from stretch sensors during wrist movements was randomly selected. These selected values were then input into the model, and the results were compared for evaluation. See Table 1.

A total of 50 stretch sensor values generated during wrist movements were randomly selected and inputted into the 3D wrist motion measurement model created in Section 3.4. Comparing the obtained angles with the actual wrist angles, the errors for each axis were 4.05637°, 4.23132°, and 4.31853°, respectively, all of which did not exceed 4.5°. The standard deviations were 2.2263, 2.5311, and 3.166, indicating an acceptable level for most motion capture requirements. Therefore, the 3D wrist angle measurement model proposed in this paper is applicable in practical motion capture.

## 4. Discussion

The developed stretch sensor-based algorithm showcased promising outcomes in measuring wrist angles across three dimensions. The initial linear regression in Section 3.1 analysis revealed substantial coefficients of determination (R values) for flexion/extension (F/E), radial/ulnar deviation (R/U), and pronation/supination (P/S) angles. The high R values of 0.8874, 0.8989, and 0.9421 indicated a commendable fit of the linear regression model to the data, affirming that sensor-based measurements captured a significant portion of the variability in wrist angles. However, it is noteworthy that the R values for F/E and R/U were comparatively less satisfactory than those for P/S. To enhance accuracy, a Multi-Layer Perceptron (MLP) regression analysis was implemented, incorporating 1000 training epochs, a learning rate of 0.1, and 10 neurons in the hidden layer. The outcomes of the MLP regression analysis, as depicted in Section 3.2, demonstrated improved fitting regression for F/E, R/U, and P/S sensors values, addressing the limitations observed in the initial linear regression. The innovation continued with the development in a 3D wrist angle measurement algorithm, integrating coupling signals during wrist angle movements. By inputting values from the stretch sensor’s three axes and the three angles associated with wrist movements into the MLP model, the algorithm successfully measured three-dimensional wrist angles. High R values of 0.93972, 0.94826, and 0.94742 for the train, validation, and test datasets indicated the model’s excellent fit for accurate three-dimensional wrist angle measurement. Validation of the 3D wrist angle measurement model using randomly selected stretch sensor values during wrist movements demonstrated errors within acceptable limits—values of 4.05637°, 4.23132°, and 4.31853° for each axis, all below 4.5°. Standard deviations of 2.2263, 2.5311, and 3.166 further affirmed the model’s applicability for practical motion capture requirements. As the experiment was repeated, a decline in the durability of the sensor became apparent. This aspect could be considered a significant drawback when used in real-life applications. Additionally, when noise due to external forces occurs, there is difficulty in distinguishing it from the coupling shown in Section 3.3. While this was not relevant in the experiments conducted in this paper due to the absence of external interference, algorithms specifically designed to handle such noise will be necessary for real-world applications. Therefore, in future research, we will investigate methods for developing algorithms to further enhance the accuracy of the sensor and explore approaches to improve the durability of the sensor.

## 5. Conclusions

In conclusion, our study successfully developed a smart wearable sensor utilizing a stretch sensor immersed in a SWCNT dispersion, enabling the measurement of complex joint movements, particularly the wrist angle. The employed stretch sensor exhibited notable advantages, including its lightweight nature, as well as enhancing wearer convenience and portability. Its compatibility with the human body, thanks to flexible and elastic materials, provided comfort and unrestricted movement, making it highly applicable in bio-signal measurement and motion analysis across various fields. The versatility and effectiveness of the stretch sensor were effectively demonstrated in the successful utilization for wrist angle measurement within this paper. Additionally, the performance of the wrist angle measurement model was enhanced through the comparison of R values using MLP (Multi-Layer Perceptron), a deep learning technique. The model, analyzing results from three sensors, successfully measured 3D wrist angles without relying on complex decoupling regression formulas used in conventional methods. Looking ahead, this precise measurement model holds significant promise in the fields of motion analysis and rehabilitation. However, it is crucial to acknowledge certain limitations, such as the observed reduction in sensor durability with repeated experiments. Addressing these issues will be a focal point for future research, aiming to further enhance the accuracy of the sensor through algorithm development and explore methods to improve its overall durability in practical applications. This study has introduced a novel framework for interpreting data generated by wearable sensors such as stretch sensors, laying the groundwork for the evolution of wearable sensors in the fields of healthcare and biomechanics where this framework can be applied.

## Figures and Tables

**Figure 1 sensors-24-01685-f001:**
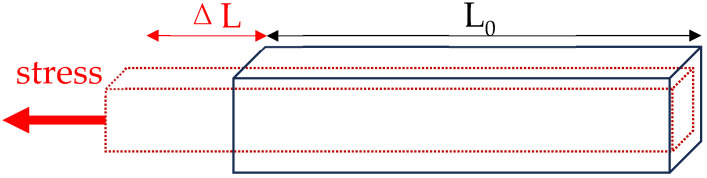
Principle of stretch sensor.

**Figure 2 sensors-24-01685-f002:**
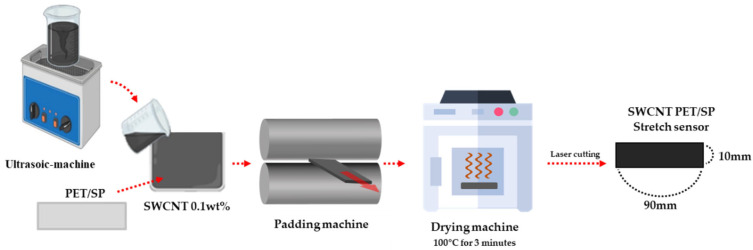
The fabrication process of the SWCNT stretch sensor.

**Figure 3 sensors-24-01685-f003:**
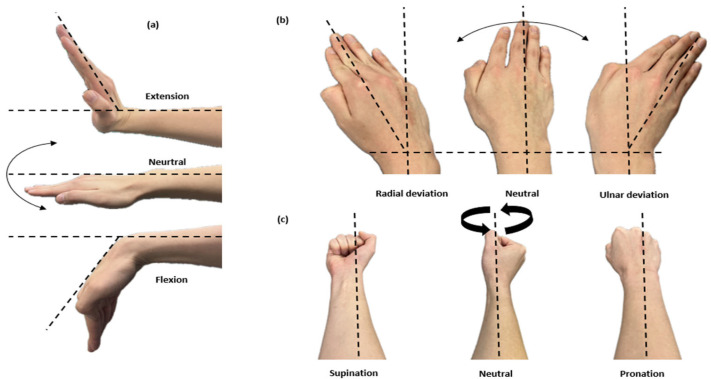
Wrist exercises: (**a**) flexion/extension (F/E); (**b**) radial/ulnar deviations (R/U); (**c**) pronation/supination (P/S).

**Figure 4 sensors-24-01685-f004:**
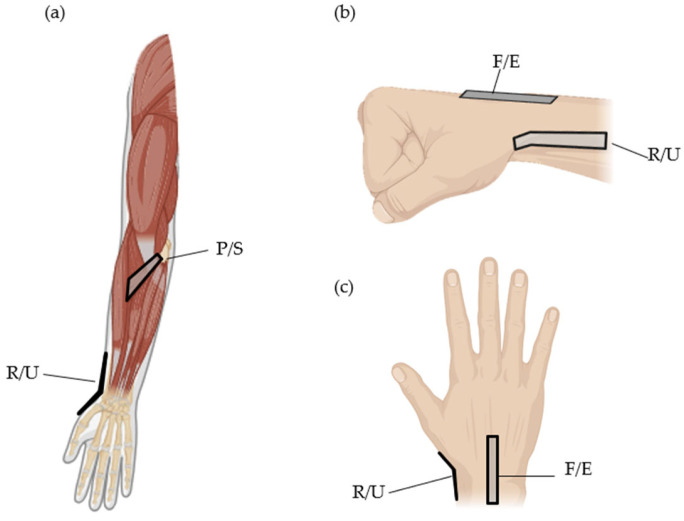
Anatomically based stretch sensor location schematic; (**a**) anterior view (**b**) lateral view (**c**) posterior view.

**Figure 5 sensors-24-01685-f005:**
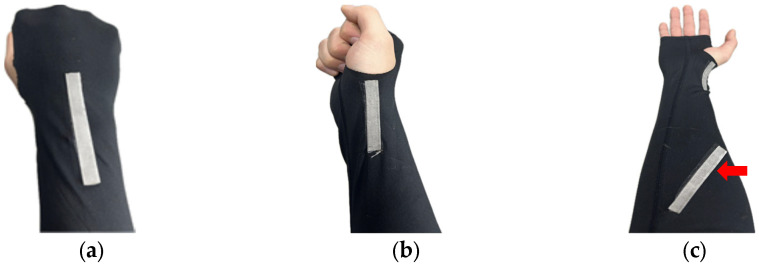
Stretch sensor placement; (**a**) flexion/extension; (**b**) radial deviation/ulnar deviation; (**c**) pronation/supination.

**Figure 6 sensors-24-01685-f006:**
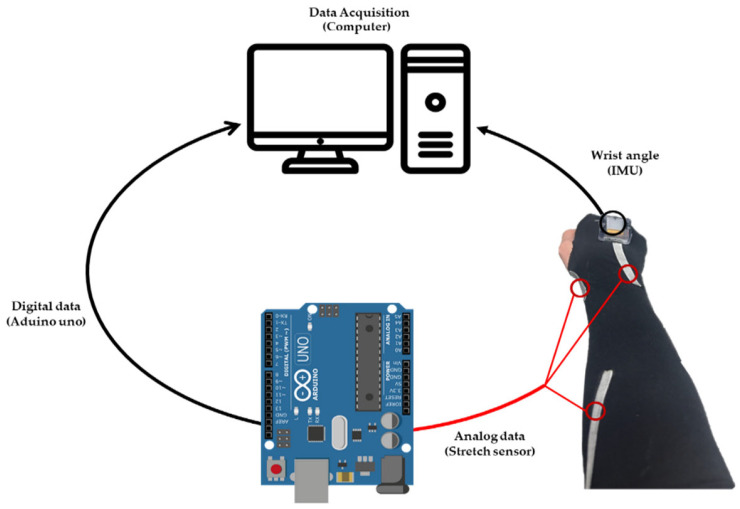
The method of obtaining data for stretch sensors and IMU.

**Figure 7 sensors-24-01685-f007:**
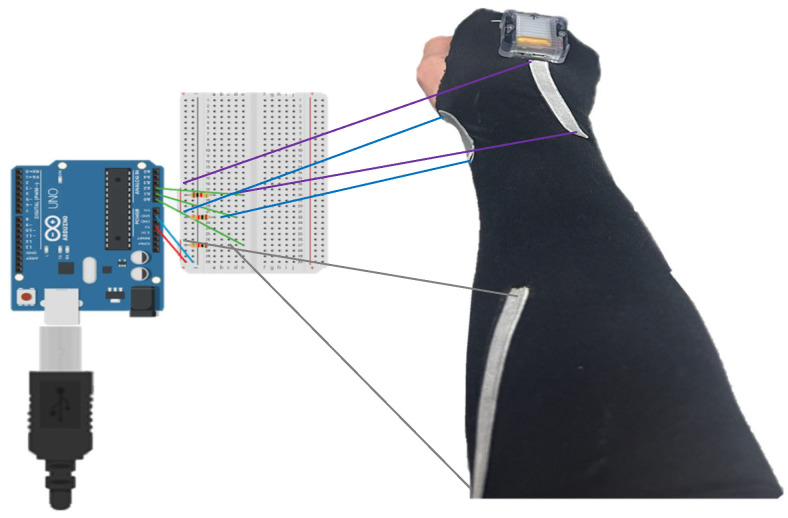
Stretch sensor and Arduino Uno Connection Circuit Diagram.

**Figure 8 sensors-24-01685-f008:**
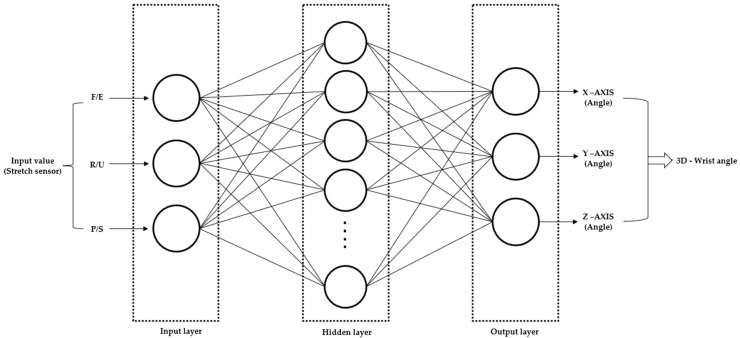
MLP Model.

**Figure 9 sensors-24-01685-f009:**
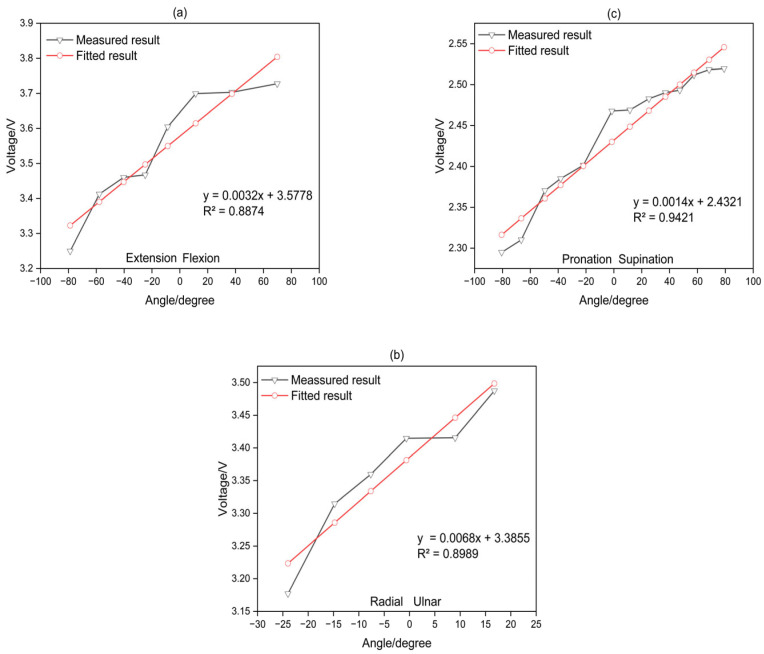
Measurements and linear fitting of wrist motion: (**a**) flexion/extension; (**b**) radial/ulnar deviation; (**c**) pronation/supination.

**Figure 10 sensors-24-01685-f010:**
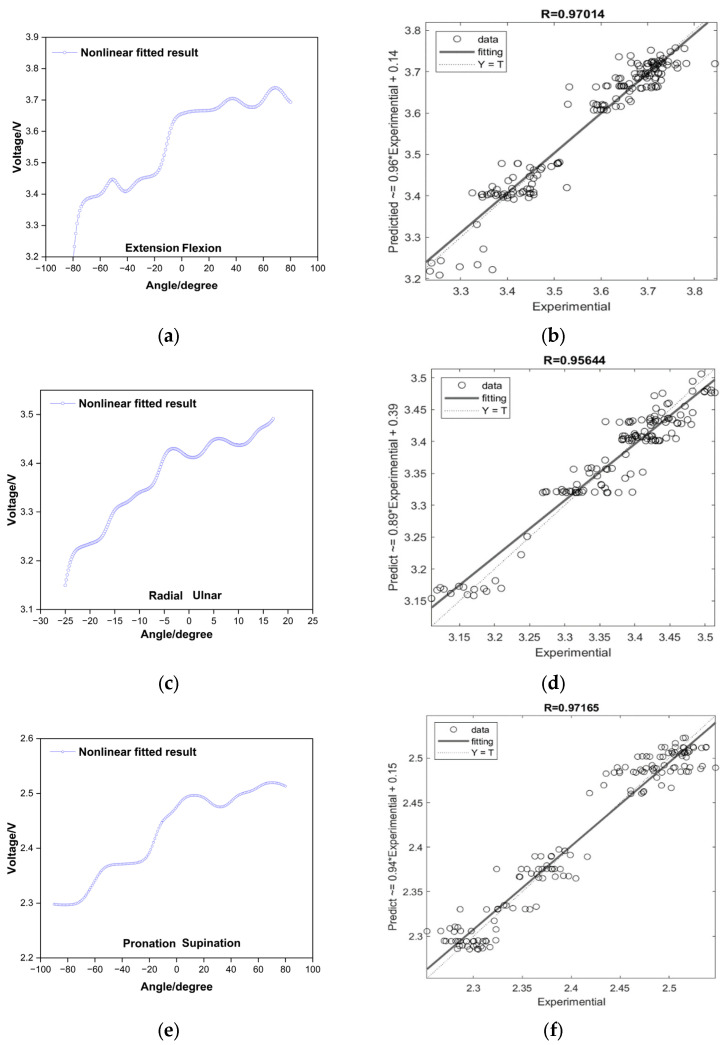
Nonlinear MLP fitting of wrist motion: (**a**) flexion/extension; (**b**) radial/ulnar deviation; (**c**) pronation/supination; (**d**) R value for F/E; (**e**) R value for R/U; (**f**) R value for P/S.

**Figure 11 sensors-24-01685-f011:**
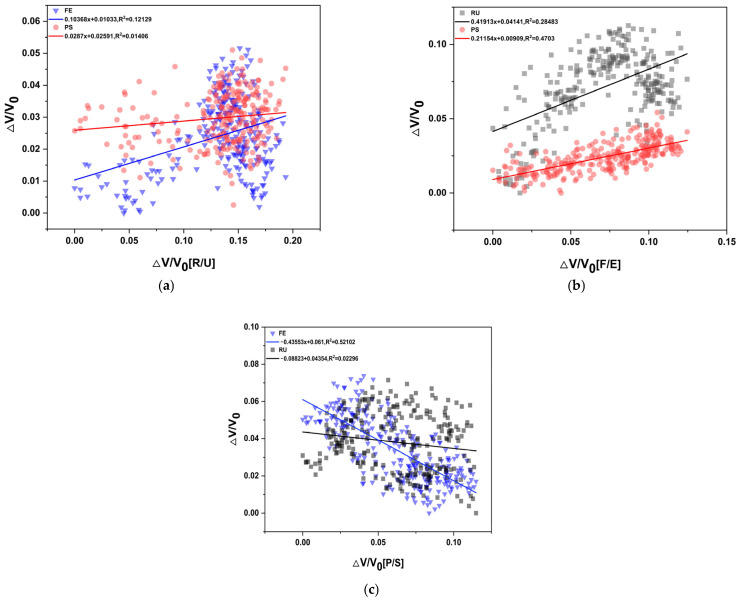
Coupling of sensors generated during wrist movements along the main axis: (**a**) F/E; (**b**) R/U; (**c**) P/S.

**Figure 12 sensors-24-01685-f012:**
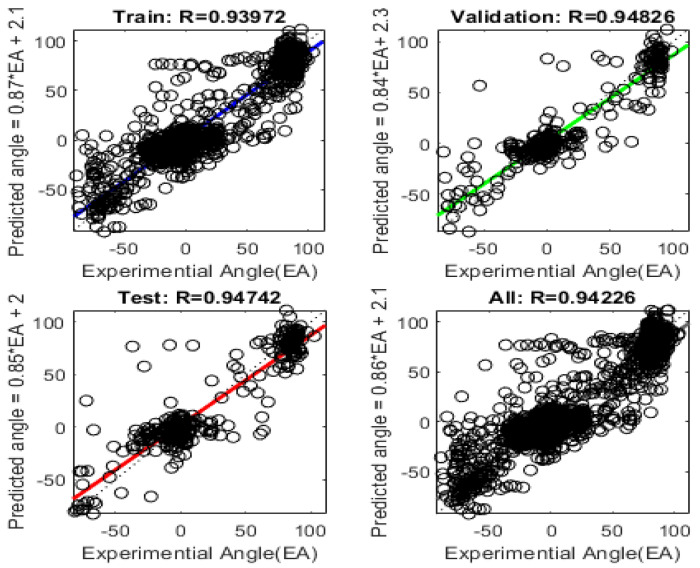
The regression analysis of the 3D wrist angle measurement model, created through MLP fitting with a 3:3 input–output configuration, was conducted.

**Figure 13 sensors-24-01685-f013:**
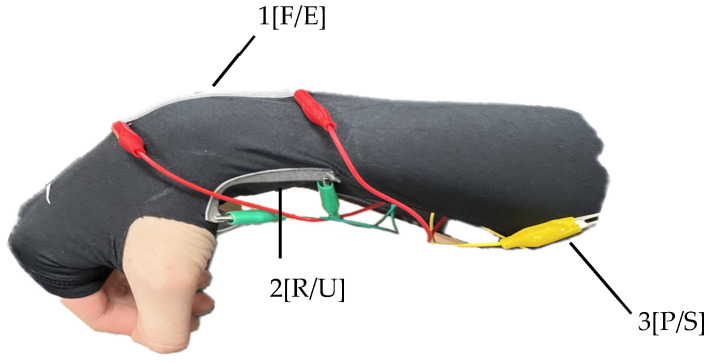
3D wrist motion measurement.

**Table 1 sensors-24-01685-t001:** At random 50 wrist angles, the three stretch sensor voltages and the error angles, as depicted in Figure 13.

	Output Voltage/V			Angular Error |Predicted Angle − Actual Angle|/Degree		
	F/E	R/U	P/S	No. 1	No. 2	No. 3
1	3.3378	3.2248	2.4724	6.57	6.23	1.57
2	3.3444	3.2537	2.4547	2.71	7.61	2.79
3	3.6414	3.2433	2.4668	7.92	3.63	4.22
4	3.7235	3.4021	2.4717	0.91	2.61	10.57
5	3.5070	3.4497	2.5022	4.71	1.69	5.88
6	3.5083	3.4040	2.4794	6.46	0.15	1.75
48	3.7188	3.4201	2.3704	6.92	3.60	0.05
49	3.7362	3.4405	2.4855	4.91	2.15	3.94
50	3.7374	3.5114	2.5121	4.69	2.62	8.14
			Mean error	4.06	4.23	4.32
			SD	2.23	2.53	3.17

## Data Availability

Data are contained within the article.

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
