# Peer review of "Measurement of 3D Wrist Angles by Combining Textile Stretch Sensors and AI Algorithm"

_sensors, 2024, doi:10.3390/s24051685_

Round 1

Reviewer 1 Report

Comments and Suggestions for Authors

The manuscript reports the use of SWCNT based resistive stretch sensors for measurement of 3D wrist angles.  Deep learning (MLP) was used to interpret sensor data for extraction of angle information.  The SWCNT stretch sensor has been widely reported previously, while its use for the target application appears a good fit and interesting. The data interpretation method is shown to work nicely.  However, the following issues should be addressed before acceptance.

1. The introduction is inadequate. Existing work on SWCNT stretch sensors is not reviewed and cited at all except one of the authors' own work. The argument on limitations of DES is also vague and inadequate (Line 49-51). 

2. Elaboration on immersion process is needed (Line 94-97). What is the difference between soaking and immersion steps, and what exactly does the padder do?

3. Figure 4 needs improvement. The photos are not adequate to show the exact locations of the sensors relative to wrist features. Maybe illustrations with wrist features and labels will work better.

4. How much does it matter to precisely place the sensors for intended measurement?

5. Figure 5 only shows minimum information. How exactly are the sensors connected to Uno, the resistors and 5V source? Circuit schematic?

6. The MLP implementation details are missing in Section 2.6, while some of them appear in Section 4 Discussion (Line 305-308), a weird arrangement. 

7. Figures 7, 8 and 9 are impossible to read with the small fonts used. 

8. Contents in Section 4 Discussion and Section 5 Conclusions seem to have be put in wrong places.  The contents in Section 4 are really meant for Conclusions, even though it is really unnecessary to repeat all the details (and data values) already presented in previous sections. Contents in Section 5 are more suitable for a Discussion section, but should be expanded to address not only advantages but also limitations of the approach used. 

Author Response

"Thank you for your dedication in reviewing our paper. Your comments have provided us with an opportunity to revisit our work. The highlighted sections in yellow correspond to your specific comments, and other colored sections indicate revisions based on comments from both you and other reviewers. We appreciate your insightful comments once again."

Please see the attachment "I have attached the revised manuscript."

  1. The introduction is inadequate. Existing work on SWCNT stretch sensors is not reviewed and cited at all except one of the authors' own work. The argument on limitations of DES is also vague and inadequate (Line 49-51). 

-> "I have added additional content to address the shortcomings in the introduction and incorporated more references."

  1. Elaboration on immersion process is needed (Line 94-97). What is the difference between soaking and immersion steps, and what exactly does the padder do?

-> "I have revised the wording for clarity where ambiguity was identified, and I have added precise explanations regarding the functionality of the padder."

  1. Figure 4 needs improvement. The photos are not adequate to show the exact locations of the sensors relative to wrist features. Maybe illustrations with wrist features and labels will work better.

-> "I found it challenging to depict the precise sensor locations with the existing figures alone. Therefore, I have created additional anatomical illustrations to provide a clearer representation of the sensor's exact positions."

  1. How much does it matter to precisely place the sensors for intended measurement?

-> "I have added content emphasizing the importance of placing sensors in precise locations."

  1. Figure 5 only shows minimum information. How exactly are the sensors connected to Uno, the resistors and 5V source? Circuit schematic?

-> "I have added a circuit schematic and incorporated more detailed information in the corresponding explanatory section."

  1. The MLP implementation details are missing in Section 2.6, while some of them appear in Section 4 Discussion (Line 305-308), a weird arrangement. 

-> "I have reviewed and added the missing content."

  1. Figures 7, 8 and 9 are impossible to read with the small fonts used. 

-> "I have improved the overall size, arrangement, and quality of the figures."

  1. Contents in Section 4 Discussion and Section 5 Conclusions seem to have be put in wrong places.  The contents in Section 4 are really meant for Conclusions, even though it is really unnecessary to repeat all the details (and data values) already presented in previous sections. Contents in Section 5 are more suitable for a Discussion section, but should be expanded to address not only advantages but also limitations of the approach used. 

-> "I have rewritten the discussion and conclusion sections, and also discussed the limitations."

Reviewer 2 Report

Comments and Suggestions for Authors

In this manuscript, authors assessed wrist angles using a textile stretch sensor system attached to an arm sleeve. To capture all three axes of the wrist, three sensors were employed. Enhancing sensor precision, the authors incorporated the multi-layer perceptron (MLP) technique, a subset of deep learning. Rather than constraining measurement values for each sensor to specific axes, the authors devised an algorithm that leverages the interconnection among sensors, enabling the measurement of wrist angles in three dimensions. Employing this algorithm, the error angle in wrist angle measurements obtained from textile stretch sensors was less than 4.5 degrees.

The paper is exciting and can be applied in many situations. There are some comments to make this work better:

  1. The novelty of this study needs to be clarified. How does the approach proposed by the authors differ from other similar approaches? Many papers have been published using stretch sensors to explore the limitations of previous soft sensors in integration drift and the need for continuous absolute position readjustment. What is your difference from the previous works and innovation? Clearly state the novelty of this study at the end of the introduction section.
  2. Motivation for the study must be given in the introduction section. What is the knowledge gap bridged by this study?
  3. The explanation of the SWCNTs needs to be longer and more detailed. Substantial detail must be given. Most importantly, the proposed work could be more precise here.
  4. The authors should have presented the related works sufficiently.
  5. The authors chose to use the training dataset, which constituted 70% of the data, while 15% was used for validation and the remaining 15% for testing. Why?
  6. Comparing the results with state-of-the-art studies needs to be considered. While the stretch sensor is claimed to have advantages such as lightweight, reproducibility, and compatibility with the human body, the paper lacks specific validation data or comparisons with existing sensors. The absence of quantitative metrics makes objectively assessing the sensor's performance challenging.

I consider that the authors need to address these issues before their paper can be published in the journal.

Comments on the Quality of English Language

Moderate editing of the English language is required. Some content needs to be more comprehensive.

Author Response

"We sincerely appreciate your thoughtful review of our manuscript. Your valuable comments have given us an invaluable chance to reassess and refine our work. Sections highlighted in blue specifically address your feedback, while other colored revisions incorporate input from both your comments and those of other reviewers. Thank you for your valuable insights once more."

Please see the attachment "I have attached the revised manuscript."

  1. The novelty of this study needs to be clarified. How does the approach proposed by the authors differ from other similar approaches? Many papers have been published using stretch sensors to explore the limitations of previous soft sensors in integration drift and the need for continuous absolute position readjustment. What is your difference from the previous works and innovation? Clearly state the novelty of this study at the end of the introduction section.
  • "I have added original content to the conclusion section to enhance the uniqueness of this paper. Additionally, in the middle of the introduction, I have included drawbacks of the 'des' sensor to emphasize the advantages of the stretch sensor."

2.Motivation for the study must be given in the introduction section. What is the knowledge gap bridged by this study?

  • "As indicated by the title, the existing methods for wrist angle measurement are static and require separate equations. Therefore, they appear to be lacking for practical use in real-life situations. To address this, I aim to create a sensor algorithm model for more delicate and dynamic motion measurement. For this purpose, I utilized deep learning technology."

The explanation of the SWCNTs needs to be longer an01d more detailed. Substantial detail must be given. Most importantly, the proposed work could be more precise here.

  • "I have reinforced the reasons for using SWCNT to make it clearer for the readers."

3,The authors should have presented the related works sufficiently.

  • "I have added reference papers related to the previous works.".

4.The authors chose to use the training dataset, which constituted 70% of the data, while 15% was used for validation and the remaining 15% for testing. Why?

  • "I have included theoretical explanations for this."

5.Comparing the results with state-of-the-art studies needs to be considered. While the stretch sensor is claimed to have advantages such as lightweight, reproducibility, and compatibility with the human body, the paper lacks specific validation data or comparisons with existing sensors. The absence of quantitative metrics makes objectively assessing the sensor's performance challenging.

-> I felt that there was a lack of references for this content, so I added citations to enhance the credibility of the advantages. However, it's essential to note that the primary focus of this paper is not on the development of the stretch sensor but rather on analyzing the data obtained through the stretch sensor. Therefore, the emphasis on lightness, reproducibility, and advantages is used to explain the reasons for utilizing the stretch sensor."

-"I have also made overall improvements to the English composition."-

Reviewer 3 Report

Comments and Suggestions for Authors

This is the well-organized research paper on the measurement of 3D wrist angles by combining textile stretch sensors and AI algorithm

This paper could be published as it is.

Comments on the Quality of English Language

minor changes required.

Author Response

"Thank you for your positive evaluation of our paper. We truly appreciate your efforts. Attached is the revised manuscript, incorporating feedback from other reviewers. Additionally, I have made some minor edits to the English. We appreciate your review and would be grateful if you could take a look."

Round 2

Reviewer 2 Report

Comments and Suggestions for Authors

In this version, the authors have added some experimental results and modifications to respond positively to my questions. Some minor error points should be considered.

1. Various definitions should be fully presented initially, and abbreviations can be subsequently used, for example, MLP.

2. MLP (Multilayer Perceptron) or Multi-Layer Perceptron (MLP) ? 

Comments on the Quality of English Language

Just a few points are required to be reviewed.

Author Response

To the reviewer.

Thank you for your meticulous feedback. We appreciate your sharp observations pointing out areas we may have overlooked. The revised content is highlighted in blue.

  1. Various definitions should be fully presented initially, and abbreviations can be subsequently used, for example, MLP.

-> Abbreviations were preemptively explained at various points throughout the paper.

  1. MLP (Multilayer Perceptron) or Multi-Layer Perceptron (MLP)?

-> The term "Multi-Layer Perceptron (MLP)" has been standardized throughout.